# Eudebeiolide B Inhibits Osteoclastogenesis and Prevents Ovariectomy-Induced Bone Loss by Regulating RANKL-Induced NF-κB, c-Fos and Calcium Signaling

**DOI:** 10.3390/ph13120468

**Published:** 2020-12-16

**Authors:** Mi-Hwa Kim, Hyung-Jin Lim, Seon Gyeong Bak, Eun-Jae Park, Hyun-Jae Jang, Seung Woong Lee, Soyoung Lee, Kang Min Lee, Sun Hee Cheong, Seung-Jae Lee, Mun-Chual Rho

**Affiliations:** 1Biological Resources Research Group, Gyeongnam Department of Environment Toxicology and Chemistry, Korea Institute of Toxicology (KIT), Jinju 52834, Korea; mihwa@krict.re.kr; 2Immunoregulatory Material Research Center, Korea Research Institute of Bioscience and Biotechnology (KRIBB), Jeongeup 56212, Korea; lhjin@kribb.re.kr (H.-J.L.); tsk9096@kribb.re.kr (S.G.B.); pej911029@kribb.re.kr (E.-J.P.); lswdoc@kribb.re.kr (S.W.L.); sylee@kribb.re.kr (S.L.); 3Natural Medicine Research Center, Korea Research Institute of Bioscience and Biotechnology (KRIBB), Cheongju 28116, Korea; water815@kribb.re.kr; 4Department of Molecular Biology, Chonbuk National University, Jeonju 54896, Korea; kmlee@jbnu.ac.kr; 5Department of Marine Bio Food Science, Chonnam National University, Yeosu 59626, Korea; sunny3843@jnu.ac.kr

**Keywords:** eudebeiolide B, osteoporosis, RANKL, OVX mouse model, calcium signal

## Abstract

Eudebeiolide B is a eudesmane-type sesquiterpenoid compound isolated from *Salvia plebeia* R. Br., and little is known about its biological activity. In this study, we investigated the effects of eudebeiolide B on osteoblast differentiation, receptor activator nuclear factor-κB ligand (RANKL)-induced osteoclastogenesis in vitro and ovariectomy-induced bone loss in vivo. Eudebeiolide B induced the expression of alkaline phosphatase (ALP) and calcium accumulation during MC3T3-E1 osteoblast differentiation. In mouse bone marrow macrophages (BMMs), eudebeiolide B suppressed RANKL-induced osteoclast differentiation of BMMs and bone resorption. Eudebeiolide B downregulated the expression of nuclear factor of activated T-cells 1 (NFATc1) and c-fos, transcription factors induced by RANKL. Moreover, eudebeiolide B attenuated the RANKL-induced expression of osteoclastogenesis-related genes, including cathepsin K (Ctsk), matrix metalloproteinase 9 (MMP9) and dendrocyte expressed seven transmembrane protein (DC-STAMP). Regarding the molecular mechanism, eudebeiolide B inhibited the phosphorylation of Akt and NF-κB p65. In addition, it downregulated the expression of cAMP response element-binding protein (CREB), Bruton’s tyrosine kinase (Btk) and phospholipase Cγ2 (PLCγ2) in RANKL-induced calcium signaling. In an ovariectomized (OVX) mouse model, intragastric injection of eudebeiolide B prevented OVX-induced bone loss, as shown by bone mineral density and contents, microarchitecture parameters and serum levels of bone turnover markers. Eudebeiolide B not only promoted osteoblast differentiation but inhibited RANKL-induced osteoclastogenesis through calcium signaling and prevented OVX-induced bone loss. Therefore, eudebeiolide B may be a new therapeutic agent for osteoclast-related diseases, including osteoporosis, rheumatoid arthritis and periodontitis.

## 1. Introduction

Osteoporosis is characterized by a weak bone structure and a decrease in bone density; it has become a major health problem [1]. Estrogen deficiency, which is observed in the postmenopausal state and as a consequence of aging, is a major contributor to osteoporosis by inducing excessive osteoclast differentiation [2]. Bone homeostasis is maintained by the balance of osteoclastic bone resorption and osteoblastic bone formation [3]. Osteoclasts are multinucleated cells derived from monocyte–macrophage lineage precursor cells [4,5]. Macrophage colony-stimulating factor (M-CSF) and receptor activator nuclear factor-κB (NF-κB) ligand (RANKL) are the two major factors involved in osteoclast differentiation. M-CSF induces the proliferation and survival of osteoclast precursor cells, whereas RANKL stimulates osteoclast differentiation and activation. RANKL expressed in activated osteoblasts binds its receptor, RANK, which is expressed on the surface of osteoclast precursors, and then activates multiple downstream signaling pathways [6]. RANKL induces the recruitment of adaptor molecules, such as TNF receptor-associated factor 6 (TRAF6). TRAF6 activates downstream signaling pathways, including NF-κB, mitogen-associated protein kinase (MAPK) extracellular-regulated kinase (ERK), Jun N-terminal kinase (JNK), and P38, Akt, activator protein 1 (AP-1) and nuclear factor of activated T-cells, cytoplasmic 1 (NFATc1) [6,7,8,9]. In addition, one of the major downstream signals involved in osteoclast differentiation is calcium signaling, which is mediated by immunoreceptor tyrosine-based activation motif (ITAM)-bearing adaptor molecules, such as Fc receptor common γ subunit (FcRγ) and DNAX-activating protein 12 (DAP12) [8,9,10,11]. ITAM phosphorylation activates Syk, which then promotes the PLCγ. Subsequently, inositol-1,4,5-triphosphate (IP_3_) is produced, and Ca^2+^ is released from the endoplasmic reticulum (ER) via IP_3_ binding to the inositol triphosphate receptor (IP_3_R), leading to calcium mobilization [10,12]. Calcium signaling is critical for the activation of Ca^2+^/calmodulin-dependent protein kinase IV (CaMKIV) and cAMP response element-binding protein (CREB), which are required for the activation of NFATc1 [9,13,14]. CREB and NFATc1 induce the expression of osteoclast-specific genes such as tartrate-resistant acid phosphatase (TRAP), MMP9, NFATc1 and Cstk [13,15].

Natural-plant-derived extracts and compounds have become the focus of studies developing therapeutic agents for human disease due to the potentially untenable side effects of pharmaceutical agents or hormone treatments. As shown in many recent reports, compounds derived from natural products are considered new therapeutic agents for bone diseases, such as rheumatoid arthritis, periodontal disease and osteoporosis [16,17,18]. According to our previous study, a *Salvia plebeia* extract inhibited RANKL-induced osteoclastogenesis and prevented ovariectomized (OVX)-induced bone loss [19]. Eudebeiolide B, which was used in the present study, is a eudesmane-type sesquiterpenoid compound isolated from a *Salvia plebeia* R. Br. extract [20]. Our previous studies were reported that various eudesmane-type sesquiterpenoid lactones, eudebeiolides A−K [20,21] were isolated from the aerial part of *Salvia plebeia*. Among these isolates, eudebeiolide B has a 3-oxoeudesman-1(2),7(11)-dien-8,12-olide skeleton with conjugated carbonyl groups at C-3 and C-7. Although eudebeiolide B showed a mild anti-inflammatory effect based on previous results, no additional biological reports were observed in the recent literature (the effects of eudebeiolide B on osteoclastogenesis have not yet been investigated). Therefore, we attempted to investigate the effect of eudebeiolide B on RANKL-induced osteoclast differentiation and elucidate the underlying molecular mechanisms. In addition, the therapeutic effects of eudebeiolide B on OVX-induced bone loss were evaluated. The inhibition of calcium signaling in osteoporosis by targeting RANKL may be a useful approach for osteoporosis-specific chemotherapy.

## 2. Results

### 2.1. Eudebeiolide B Inhibits RANKL-Induced Osteoclast Differentiation and Function

In a previous study, we isolated various phytochemical compounds from *Salvia plebeian.* Among these compounds, sesquiterpenoids showed superior antiosteoclastogenesis activities [19]. Eudebeiolide B, which is a type of sesquiterpenoid, showed potent antiosteoclastogenesis activity and no cytotoxicity. Furthermore, the eudebiolide B content of the ethanol extract of *Salvia plebeian* was higher than other sesquiterpenoids (Appendix A) [20]. Eudebeiolide B, which has a maximum absorption wavelength of approximately 218 nm, showed a clear peak at 21.3 min on the chromatogram of the *Salvia plebeia* EtOH extract (Figure 1A and Appendix A). We examined the cell viability of BMMs using the XTT assay to confirm whether the inhibitory effects of eudebeiolide B were due to cytotoxicity. As shown in Figure 1B, cell viability was not affected by treatment with eudebeiolide B. In addition, we investigated the effects of eudebeiolide B on osteoclast differentiation and function. BMMs were pretreated with the indicated concentrations of eudebeiolide B and treated with M-CSF and RANKL to determine whether eudebeiolide B inhibited osteoclastogenesis. After four days of RANKL stimulation, TRAP-positive multinucleated osteoclasts were observed, and treatment with eudebeiolide B inhibited the formation of TRAP-positive multinucleated osteoclasts in a dose-dependent manner (Figure 1C).

Mature osteoclasts were cultured on hydroxyapatite-coated plates and treated with or without eudebeiolide B in the presence of RANKL to determine the effect of eudebeiolide B on bone resorption. Because 1 μM of eudebeiolide B could not significantly reduce osteoclast differentiation, we used 5, 10 and 30 μM of eudebeiolide B in the bone resorption assay. After 48 h, resorbed areas were generated by the mature osteoclasts. Eudebeiolide B inhibited bone resorption in a dose-dependent manner (Figure 1D). Furthermore, we investigated the effect of eudebeiolide B on osteoblast differentiation. The 30 μM of eudebeiolide B showed cytotoxicity in 21 days culture (Appendix A). Thus we investigated 1 and 10 μM of eudebeiolide B. The treatment of eudebeiolide B increased ALP activities in MC3TC-E1 murine osteoblastic cells and upregulated mineralization of MC3T3-E1 (Figure 1E,F). Furthermore, osteoblast-differentiation-related genes such as runt-related transcription factor 2 (Runx2), Osterix and osteoprotegerin (OPG) were upregulated and RANKL was downregulated by eudebeiolide B treatment (Figure 1G).

### 2.2. Eudebeiolide B Inhibits the Expression of Osteoclastogenesis-Related Marker Genes

NFATc1 forms a complex with AP-1 (Fos/Jun), PU.1 and microphthalmia-associated transcription factor (MITF), an essential transcription factor in osteoclastogenesis, for the efficient induction of osteoclast-specific genes [6,8]. We expected that eudebeiolide B would inhibit the expression of osteoclast-specific genes because eudebeiolide B suppressed RANKL-induced c-fos and NFATc1 protein expression (Figure 2A–C). As shown in Figure 2, the expression of NFATc1, Cathepsin K (Ctsk), MMP9 and DC-STAMP mRNAs was upregulated by RANKL, whereas the expression of these genes was significantly inhibited by eudebeiolide B treatment.

### 2.3. Eudebeiolide B Inhibits RANKL-Mediated Cellular Signaling

BMMs were pretreated with eudebeiolide B and stimulated with RANKL at the indicated times to investigate the mechanism by which eudebeiolide B inhibited RANKL-mediated cellular signaling. As shown in Figure 3, the phosphorylation of ERK, P38 and JNK increased in response to RANKL, but eudebeiolide B did not affect the phosphorylation of these targets. However, Akt and NF-κB p65 phosphorylation was suppressed by eudebeiolide B in the RANKL-treated BMMs. RANKL-RANK signaling activates PLCγ2 and increases the intracellular calcium levels via ITAM-harboring molecules, such as DAP12 and FcRγ, subsequently activating CaMKIV, which primarily activates CREB [12,13,14]. We examined whether eudebeiolide B affected calcium signaling induced by RANKL stimulation. Eudebeiolide B inhibited PLCγ2 and Btk activation upon RANKL stimulation. Moreover, CaMKIV/CREB, downstream signaling molecules of PLCγ2, were inhibited by eudebeiolide B treatment (Figure 4). Thus, eudebeiolide B inhibited osteoclastogenesis by suppressing RANKL-mediated calcium signaling.

### 2.4. Therapeutic Effect of Eudebeiolide B on OVX-Induced Bone Loss

We used the OVX mouse model to mimic menopause-induced bone loss in women and determine whether eudebeiolide B prevented OVX-induced bone loss. In this study, we used alendronate as a positive control because it is used as a therapeutic agent for osteoporosis. The mice were gastric eudebeiolide B, with alendronate or PBS used six weeks after the OVX or sham operation to evaluate the therapeutic effect. Dosages of eudebeiolide B were determined based on a previous study of *Salvia plebeian* R. Br [19]. We obtained 24.5 mg of eudebeiolide B from 2.6 kg of ethanol extract. Thus, 500 mg/kg of *Salvia plebeian* R. Br. ethanol extract dosage, which is used in a previous study, contained almost 5 mg/kg eudebeiolide B. Furthermore, we added 10 mg/kg of eudebeiolide B to show a significant therapeutic effect because the extract contains other active compounds. After six weeks, the mice were sacrificed, and histomorphometric parameters were analyzed. The OVX mice, but not the sham mice, exhibited atrophy and a reduced uterus weight (data not shown). Eudebeiolide B administration prevented bone mineral density (BMD) loss and bone mineral content (BMC) loss compared to the OVX mice (Figure 5A,B). A stronger protective effect was observed with the 10 mg/kg dose than with the 5 mg/kg dose of eudebeiolide B. We next analyzed the three-dimensional visualization of the femoral metaphysis using the micro-CT and histology of the distal tibiae by H&E staining, TRAP histochemical staining of TRAP and immunohistochemical staining of ALP. The three-dimensional visualization of the proximal femurs and H&E staining of tibiae revealed a loss of trabeculae and the discontinuity of cancellous bone in OVX mice, but these structures were preserved in the sham mice. Moreover, the trabecular structure was well maintained in OVX mice treated with eudebeiolide B (Figure 6 and Figure 7A). In the analysis of the microarchitectural parameters, the OVX mice exhibited decreased bone volume/tissue volume (BV/TV), trabecular number (Tb.N.) and trabecular thickness (Tb.Th.) compared with those of the sham mice, but these parameters were preserved in OVX mice treated with eudebeiolide B. Moreover, the serum levels of alkaline phosphatase (ALP), a bone formation marker, were decreased and c-terminal telopeptide of type 1 collagen (CTX), bone turnover markers, were increased in OVX mice compared with those in the sham mice. However, the serum levels of ALP were increased and CTX were decreased in the OVX mice treated with eudebeiolide B (Figure 5C,D). The histological results showed OVX operation increased TRAP activities and decreased ALP activities. However, eudebeiolide B treatment recovered TRAP and ALP activities (Figure 7B–E). In addition, the serum levels of osteoprotegerin (OPG), which is a blocker of RANKL and released from osteoblasts, were decreased and the RANKL and RANKL/OPG ratio increased in OVX mice compared with those in the sham mice. However, the treatment of eudebiolide B recovered them (Figure 5E–G). Based on these results, eudebeiolide B prevented OVX-induced bone loss.

## 3. Discussion

New therapeutic agents for osteoporosis are required to effectively regulate osteoclast and osteoblast differentiation and activity with no undesirable side effects [22,23,24,25]. Thus, the ideal strategy is the development of an antiosteoporotic agent to inhibit bone resorption and enhance bone formation. To develop this strategy, we examined natural-product-derived compounds as new phytomedicine antiosteoporotic candidates in this study. Eudebeiolide B is a eudesmane-type sesquiterpenoid compound that was isolated from *Salvia plebeian* [20]. We examined the effects of eudebeiolide B on osteoblast formation. Eudebiolide B promoted osteoblast differentiation and calcium accumulations. We investigated the effect of eudebeiolide B on osteoclastogenesis and OVX-induced bone loss. Eudebeiolide B inhibited RANKL-induced osteoclast differentiation and bone resorption through the suppression of PLCγ2, NFATc1 and c-Fos. Calcium signaling activates FcRγ and DAP12 by interacting with ITAM sequences and subsequently phosphorylating PLCγ [13,14,15]. PLCγ2 plays an important role in the adhesion, spreading and migration of preosteoclasts [26]. Additionally, targeted deletion of PLCγ2 resulted in an osteopetrotic phenotype in vivo and decreased RANKL-mediated NFATc1 expression in vitro [25,27]. CaMKIV/CREB induce c-Fos expression and NFATc1 amplification, resulting in the transcription of osteoclast-specific genes [13,28]. CaMKIV knockout mice exhibited increased bone mineral density and decreased bone resorption [13]. Binding of RANKL to RANK activates MAPKs and NF-κB signaling through and calcium signaling. If eudebeiolide B inhibited the receptor directly, downstream signaling pathways including MAPKs, NF-κB and calcium signaling were inhibited by eudebeiolide B treatment. MAPK’s signaling was not affected by eudebeiolide B treatment. These results imply eudebeiolide B exerts its activity by directly inhibiting specific signaling molecules. NFATc1 regulates the expression of a number of osteoclast-specific genes, such as TRAP, MMP9, cathepsin K and DC-STAMP, which are important for osteoclast differentiation and function [7,8,29,30,31]. These genes are upregulated during RANKL-induced osteoclastogenesis and attenuated by eudebeiolide B treatment, suggesting that eudebeiolide B not only affects the expression of NFATc1 but also the expression of its downstream genes.

Estrogen deficiency is related to the increase in bone turnover, which accelerates bone loss and eventually increases the risk of osteoporosis [2]. Therefore, a postmenopausal mouse model established by ovariectomy has been widely used for in vivo screens of new drug candidates [32]. OVX increased bone turnover, as shown by the decrease in serum ALP and OPG and the increased osteocalcin, CTX and RANKL levels, but the serum levels of these markers were recovered by the administration of eudebeiolide B in OVX mice, indicating that eudebeiolide B prevents bone loss by inhibiting bone turnover.

## 4. Materials and Methods

### 4.1. Reagents

All cell culture reagents, including media, antibiotics and fetal bovine serum, were purchased from Gibco BRL (Grand Island, NY, USA). Recombinant human M-CSF and human RANKL were purchased from Peprotech (London, UK). Specific antibodies against p-ERK (Tyr202/204), p-P38, p-JNK (Tyr1007/1008), p-Akt, p-CREB, p-PLCγ2 (Tyr759), p-Btk (Tyr223), NFATc1 and c-Fos were obtained from Cell Signaling Technology (Boston, MA, USA). The monoclonal β-actin antibody was purchased from Sigma (St. Louis, MO, USA), and secondary antibodies were obtained from Santa Cruz Biotechnology (Santa Cruz, CA, USA).

### 4.2. Isolation of Eudebeiolide B

After preparing the ethanolic *Salvia plebeia* R. Br. extract, the residue (2.0 kg) was suspended in distilled water (2 L) and then progressively partitioned using n-hexane (30 L) and ethyl acetate (EtOAc) (30 L) to yield the n-hexane- (779 g) and EtOAc-soluble fractions (541 g), respectively. The EtOAc-soluble fraction was subjected to silica gel column chromatography and eluted with a stepwise gradient of increasing concentrations of CH_3_OH in CHCl_3_ (100:0, 50:1, 25:1, 10:1, 5:1, 2:1, 1:1 and 0:1) to obtain 14 subfractions (SPE1–SPE14) that were combined for thin-layer chromatography (TLC) analysis. SPE6 (9.5 g) was loaded onto a silica gel column to yield 15 additional subfractions (SPE6A–SPE6O) using a medium-pressure liquid chromatography (MPLC) gradient solvent system (CHCl_3_:CH_3_OH, 1:0/0:1, *v*/*v*), and SPE6D (1.39 g) was applied to the C18 MPLC column to generate 19 fractions (SPE6D1–SPE6D19) with a gradient solvent system composed of H_2_O:MeOH (9:1/0:1, *v*/*v*). Eudebeiolide B (24.5 mg) was obtained from SPE6D6 (0.12 g) by semipreparative HPLC (CH_3_CN:H_2_O, 25:75, *v*/*v*, 6 mL/min). For in vitro analysis, eudebeiolide B was dissolved in dimethyl sulfoxide (DMSO) (50 μM) and treated with 1 μL of serially diluted eudebeiolide B. For in vivo analysis, eudebeiolide B was suspended in 2% carboxymethyl cellulose (CMC) and administered.

### 4.3. Determination of Eudebeiolide B Content in Salvia Plebeia EtOH Extract

For the HPLC-DAD analysis, the Agilent 1200 HPLC system (Agilent Technologies, Wilmington, DE, USA), consisting of a quaternary pump (G1311A), a degasser (G1322A), an autosampler (G1329A), a column oven (G1316A) and a diode array detector (G1315D) at a 190–400 nm wavelength, was employed. A reverse-phase column (J′sphere ODS-H80, 4 μm, 4.6 × 150 mm, YMC, Kyoto, Japan) maintained at 25 °C was used, and elution was performed with a gradient of water (A, H_2_O) and acetonitrile (B, CH_3_CN) at a flow rate of 1 mL/min. The following solvent gradient was used: 15% B within 5 min; from 15% to 40% B within 35 min; from 40% to 100% B within 5 min. The quantification of eudebeiolide B was performed using a calibration curve at six concentrations over the linear range (0.5–100.0 μg/mL).

### 4.4. Cell Culture

Mouse osteoblastic MC3T3-E1 cells were purchased from ATCC (Rockville, MD, USA) and cultured in α-MEM containing 10% FBS, 100 U/mL penicillin and 100 mg/mL streptomycin (Gibco). For differentiation, the medium was added to 50 mg/mL ascorbic acid and 10 mM β–glycerophosphate (Sigma-Aldrich, St. Louis, MO, USA). Mouse bone marrow cells were obtained from 5-week-old ICR mice and differentiated to bone marrow macrophages (BMMs) in α-MEM containing 10% FBS, 30 ng/mL M-CSF, 100 U/mL penicillin and 100 mg/mL streptomycin.

### 4.5. In Vitro Osteoclastogenesis Assay

Mouse bone marrow cells obtained from 5-week-old ICR mice were cultured as previously described [19]. Briefly, BMMs were plated on 48-well culture plates at a density of 2 × 10^4^ cells/well and treated with M-CSF (30 ng/mL) and RANKL (100 ng/mL) after pretreatment with various concentrations of eudebeiolide B. DMSO and PBS were used as vehicles of eudebeiolide B and M-CSF, RANKL. After 5 days, the cells were fixed and stained with TRAP according to the manufacturer’s instructions (Sigma-Aldrich). In brief, cells were fixed with 10% formaldehyde and washed with deionized water. The sample was then stained with a TRAP solution for 1 h. TRAP solution was prepared following the manufacturer’s instructions. After staining, cells were washed and TRAP-positive multinucleated osteoclasts were counted.

### 4.6. Cell Viability Assay

BMMs were seeded in 96-well plates at a density of 1 × 10^4^ cells/well. The cells were treated with various concentrations of eudebeiolide B in the presence of M-CSF (30 ng/mL) for 72 h. DMSO and PBS were used as vehicles of eudebeiolide B and M-CSF. A 50 µL volume of XTT solution (Cell Signaling Technology) was then added to the culture plate and incubated for 3 h. The absorbance was measured at 450 nm using a microplate ELISA reader (Molecular Devices, Sunnyvale, CA, USA).

### 4.7. Bone Resorption Assay

BMMs were seeded on hydroxyapatite-coated plates (Corning, NY, USA) at a density of 2 × 10^4^ cells/well and cultured with M-CSF (30 ng/mL) and RANKL (50 ng/mL) for 3 days. Mature multinucleated osteoclasts were treated with or without eudebeiolide B for 48 h. DMSO and PBS were used as vehicles of eudebeiolide B and M-CSF, RANKL. After incubation, a 1.2% sodium hypochlorite solution was added to each well to promote cell detachment. The plate was rinsed with distilled water, and the resorption pits were analyzed using the ImageJ program.

### 4.8. Alkaline Phosphatease (ALP) Staining and Determination of ALP Activity

MC3T3-E1 cells were seeded at 1 × 10^4^ cells/well in a 12-well plate and incubated for 2 days. The media was changed with differentiation media, and cells were treated with eudebeiolide B for 7 days. For the ALP staining, cells were fixed with 10% formaldehyde and stained with BCIP/NBT substrate solution (Sigma-Aldrich). For the measuring ALP activity, cells were lysed with lysis buffer and centrifuged. The supernatants were analyzed by an alkaline phosphatase assay kit (Abcam, Cambridge, MA, USA) following the manufacturer’s instructions. The ALP activity was normalized by total protein concentrations.

### 4.9. Alizarin Red Staining

MC3T3-E1 cells were seeded at 1 × 10^4^ cells/well in a 12-well plate and incubated for 2 days. The media was changed with differentiation media, and cells were treated with eudebeiolide B for 21 days. The medium was changed every 3 days. The cells were then washed with PBS and fixed with 10% formalin for 30 min at room temperature. After fixation, the cells were washed with PBS and stained with 2% Alizarin red S staining solution (Sigma-Aldrich) for 45 min at room temperature in the dark. The cells were washed with distilled water and dissolved with 10% (*w*/*v*) cetylpyridinium chloride in 10 mM sodium phosphate. The absorbance was measured at 562 nm.

### 4.10. Western Blot Analysis

BMMs were pretreated with eudebeiolide B for 1 h and then stimulated with RANKL (100 ng/mL) for the indicated times. DMSO and PBS were used as vehicles of eudebeiolide B and M-CSF, RANKL. Total protein was extracted using a cell lysis buffer (Cell Signaling Technology) containing a protease and phosphatase inhibitor cocktail (Thermo Scientific, Cheshire, UK). Protein concentrations were measured using a bicinchoninic acid protein assay kit (Sigma-Aldrich Co., St. Louis, MO, USA). Equal amounts of protein were separated on 4–12% SDS-PAGE gels and transferred onto PVDF membranes (Amersham Bioscience, Freiburg, Germany). The membranes were blocked with 1× TBS containing 5% skim milk and washed with 1× TBS containing 0.1% Tween-20 (TBST). After blocking, the membranes were incubated with the appropriate primary antibodies and washed with TBST. After washing. The membranes were incubated with the appropriate secondary antibodies and washed with TBST. The antibodies were diluted to 1:1000 in TBS containing 5% BSA. Finally, target-specific signals were detected using an enhanced chemiluminescence solution (Intron Biotechnology, Seongnam, Korea).

### 4.11. Quantitative Real-Time RT-PCR

Total RNA was extracted from the total cell lysates with a PureLink RNA Mini kit (Invitrogen, San Diego, CA, USA), according to the manufacturer’s instructions. The complementary DNA was synthesized from 1 µg/mL of total RNA using SuperScript III First-Strand Synthesis System for RT-PCR (Thermo Scientific). Real-time PCR was performed using the StepOnePlus Real-Time PCR System using the TaqMan probe with the TaqMan Real-Time PCR Master Mixes (Applied Biosystems, Foster City, CA, USA). The relative mRNA expression levels were quantified using the threshold cycle (Ct) value, and the Ct value was normalized using the levels of the mouse GAPDH gene as an endogenous reference.

### 4.12. In Vivo Models

Six-week-old female C57BL/6 mice were purchased from OrientBio (Seongnam, Korea). The mice were randomly divided into 5 groups (*n* = 6 mice per group): sham-operated mice (SHAM), OVX mice treated with vehicle (OVX), OVX mice treated with 1 mg/kg alendronate (ALN) and OVX mice treated with eudebeiolide B (5 mg/kg or 10 mg/kg). Alendronate was used as a positive control. After stabilization for one week, the mice received a sham operation or a bilateral ovariectomy (OVX) under anesthesia. The mice were maintained without any treatment for 6 weeks after surgery until bone loss was apparent. After 6 weeks, eudebeiolide B and alendronate were suspended in 2% CMC and intragastrically injected into the appropriate groups once daily for 6 weeks. The animal experimental protocols were approved by the Institutional Animal Care and Use Committee of Korea Research Institute of Bioscience and Biotechnology (permission number # KRIBB-AEC-17059) and all mice were handled in accordance with the Guide for the Care and Use of Laboratory Animals published by the US National Institutes of Health.

### 4.13. Dual-Energy X-Ray Absorptiometry (DEXA)

At the end of the experiment, the mice were euthanized, and the bone mineral density (BMD) and bone mineral content (BMC) of the total body area were analyzed using GE Lunar PIXImus2 Dual-energy X-ray absorptiometry (GE Healthcare, Madison, WI, USA), according to the manufacturer’s instructions. The instrument was calibrated using a phantom provided by the manufacturer.

### 4.14. Microcomputed Tomography (CT) Measurements

For the micro-CT analysis, the femurs were removed from the mice, cleaned of soft tissue, fixed in 10% formalin and stored in PBS at 4 °C. The bone morphometric parameters and microarchitectural properties of the femurs were analyzed using a SkyScan 1076 micro-CT scanner (Bruker microCT, Kontich, Belgium). The femurs were prepared in polystyrene tubes filled with PBS to prevent drying during scanning. The specimen was mounted on the 1076 scanner sample chamber for micro-CT imaging and the sample was rotated automatically along the axis (angular step: 0.42°, reconstruction angular range: 360.36°). Scanning was performed using a 0.5 mm Al filter, an 88 kV source voltage and a 112 μA source current with a 9 μm resolution. The region of interest was defined between 0.5 mm and 2.5 mm below the growth plate of the proximal femur. The bone volume per tissue volume (BV/TV), trabecular number (Tb.N.), trabecular thickness (Tb.Th.) and three-dimensional images of the femur were analyzed using the Nrecon^®^, CTAn^®^ and CTVol^®^ software programs.

### 4.15. Biochemical Analysis of Serum

Blood samples were collected from all mice by cardiac puncture before sacrifice. Serum was obtained by centrifugation and then stored at −80 °C for the analysis of biochemical parameters. The serum levels of ALP, CTX, OPG and RANKL were measured by commercial ELISA kits following the manufacturer’s instructions. The ELISA kits of ALP, OPG and RANKL were obtained from Abcam and CTX was obtained from immunodiagnostic systems (Boldon, UK).

### 4.16. Histological Analysis

For histological analysis, the tibiae were removed from the mice, cleaned of soft tissue, fixed in 10% formalin for a day and decalcified in 10% ethylenediaminetetraacetic acid (EDTA) at 4 °C for a month. The decalcified sample was embedded in paraffin, sectioned at 5 μm and deparaffinized. H&E staining and histochemical staining of TRAP were then performed using an acid phosphatase staining kit (Sigma-Aldrich) according to the manufacturer’s instructions. Immunohistochemical staining for ALP was performed with the following procedures. In brief, deparaffinized slide was subjected to heat-induced antigen retrieval in 1 mM EDTA (pH8) for 15 min and incubated for 10 min in 3% hydrogen peroxide and blocked with TBST containing 5% normal goat serum for 1 h at room temperature. After blocking, the anti-ALP primary antibody (Abcam, Cambridge, UK) was bound overnight at 4 °C and washed with TBST, incubated on a slide with HRP-conjugated secondary antibody, developed with diaminobenzidine (DAB) substrate and counter-stained with hematoxylin. The osteoclast surface/bone surface and osteoblast surface/bone surface were measured using HistoMorph software and ImageJ software.

### 4.17. Statistical Analysis

Statistical analyses were performed on data collected in triplicate for all the experiments. All quantitative results are presented as means ± standard deviations (SD). Statistical analyses were performed using Prism 5 software (GraphPad Software, San Diego, CA, USA), and statistical significance was determined by one-way ANOVA followed by Dunnett’s test or the unpaired Student’s *t*-test.

## 5. Conclusions

This study was a primary investigation of eudebeiolide B, which was shown to promote osteoblast formation and inhibit osteoclastogenesis and function in vitro as well as bone loss in vivo. We found that eudebeiolide B inhibition was mediated by the downregulation of calcium signaling, NFATc1 and c-Fos induced by RANKL stimulation. These results will improve our understanding of the influences of biological activities on eudebeiolide B. Eudebeiolide B is a promising therapeutic candidate for treating osteoblast-related diseases and osteoclast-related diseases, such as osteoporosis, an important factor in aging. Although this study found the effects of the biological activities of eudebeiolide B natural compounds, there is clearly an urgent need for more research into the correlations between structural changes or bioavailability and influence factors.

## Figures and Tables

**Figure 1 pharmaceuticals-13-00468-f001:**
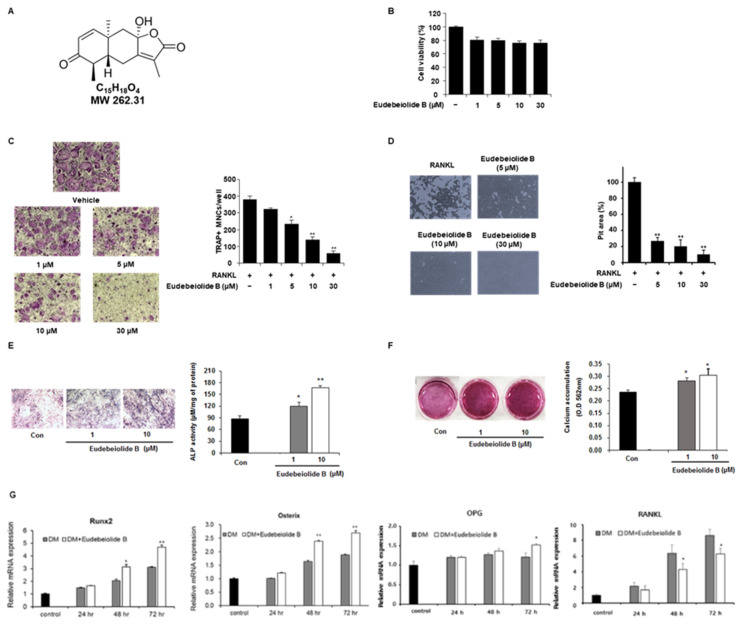
Eudebeiolide B inhibits RANKL-induced osteoclast differentiation and promotes osteoblast differentiation. (**A**) Structure of eudebeiolide B (*n* = 3). (**B**) Cytotoxicity of eudebiolide B. Bone marrow macrophages (BMMs) were seeded and treated with indicated concentrations of eudebeiolide B in the presence of M-CSF (30 ng/mL) for 72 h. Cell viability was analyzed using the XTT assay (*n* = 3). (**C**) BMMs were treated with RANKL (100 ng/mL) and M-CSF (30 ng/mL) after pretreatment with 1, 5, 10 or 30 μM of eudebeiolide B for 1 h. Cells were fixed and stained with TRAP staining solution. TRAP-positive multinucleated cells (TRAP + MNCs) with more than three nuclei were defined as osteoclasts and counted (*n* = 3). (**D**) BMMs were seeded and treated with RANKL (50 ng/mL) and M-CSF (30 ng/mL) for 72 h. Cells were then treated with eudebeiolide B for 48 h. Bone resorption areas were measured using ImageJ (*n* = 3). (**E**) MC3T3-E1 cells were seeded and incubated with differentiation media and eudebeiolide B for 7 days. Osteoblast differentiation was determined by ALP staining, and ALP activity was measured using the cell lysate (*n* = 3). (**F**) MC3T3-E1 cells were seeded and incubated with differentiation media and eudebeiolide B for 21 days. Osteoblast differentiation was assessed by Alizarin red staining, and calcium accumulation was quantified with cetylpyridinium chloride solution (*n* = 3). (**G**) MC3T3-E1 cells were treated with or without eudebeiolide B (10 μM) and cultured for 24, 48 and 72 h. The mRNA expression of Runx2, Osterix, OPG and RANKL was assessed by quantitative PCR (*n* = 3). Values are expressed as the means ± S.D. of three individual experiments. * *p* < 0.05 and ** *p* < 0.01 versus control (**C**–**E**) obtained through one-way ANOVA followed by Dunnett’s test; * *p* < 0.05 and ** *p* < 0.01 versus the only-DM-treated group (**G**) obtained through the unpaired Student’s *t*-test.

**Figure 2 pharmaceuticals-13-00468-f002:**
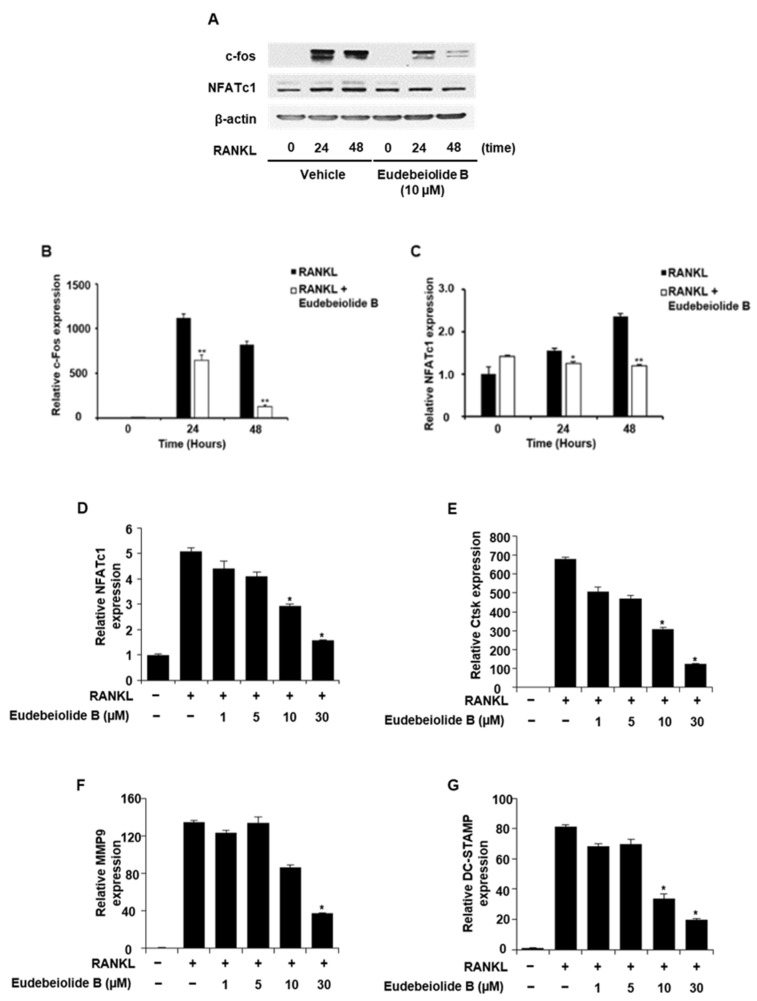
Attenuation of RANKL-induced osteoclastogenesis-related transcription factor and gene expression by eudebeiolide B. (**A**–**C**) BMMs were pretreated with or without eudebeiolide B and stimulated with RANKL (100 ng/mL) for the indicated time. (**A**) Western blot analysis of c-Fos and NFATc1 protein was performed, and band densities of (**B**) c-Fos and (**C**) NFATc1 were quantified by ImageJ (*n* = 3). (**D**–**G**) BMMs were cultured in the presence of RANKL (100 ng/mL) with or without eudebeiolide B for 48 h. (**D**) NFATc1, (**E**) cathepsin K, (**F**) MMP9 and (**G**) DC-STAMP mRNAs were analyzed by quantitative RT-PCR (*n* = 3). All data are expressed as the means ± S.D. of three individual experiments. * *p* < 0.05 and ** *p* < 0.01 versus the only-RANKL-treated group (**B**,**C**) obtained through the unpaired Student’s *t*-test; * *p* < 0.05 and ** *p* < 0.01 versus the only-RANKL-treated group (**D**–**G**) obtained through one-way ANOVA followed by Dunnett’s test.

**Figure 3 pharmaceuticals-13-00468-f003:**
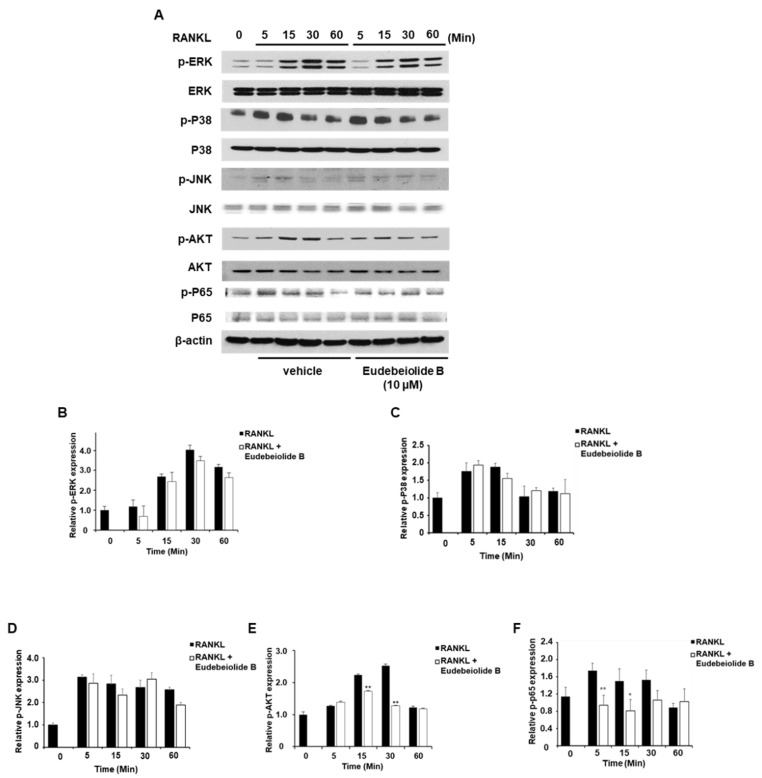
Effect of eudebeiolide B on RANKL-induced cellular signaling. BMM cells were pretreated with or without eudebeiolide B (10 μM) for 1 h and then stimulated with RANKL (100 ng/mL) for the indicated times. (**A**) Expression of signaling molecules determined by Western blot analysis. Band optical densities of (**B**) ERK, (**C**) P38, (**D**) JNK, I AKT and (**F**) NF-κB p65 were evaluated by ImageJ software (*n* = 3). All data are expressed as the means ± S.D. * *p* < 0.05 and ** *p* < 0.01 versus the only-RANKL-treated group obtained through the unpaired Student’s *t*-test.

**Figure 4 pharmaceuticals-13-00468-f004:**
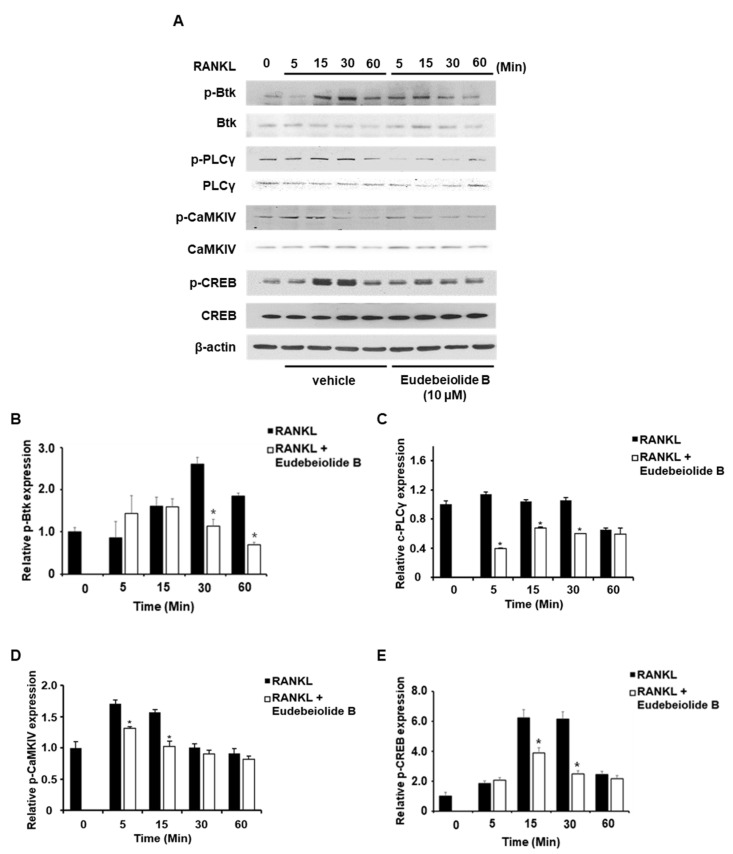
Eudebeiolide B inhibits RANKL-induced calcium signaling. (**A**) Phosphorylation was evaluated by Western blot analysis. Eudebeiolide B inhibits the phosphorylation of (**B**) Btk, (**C**) PLCγ2, (**D**) CaMKIV and (**E**) CREB. BMMs were pretreated with or without eudebeiolide B (10 μM) for 1 h and then stimulated with RANKL (100 ng/mL) for the indicated times. Western blot analyses were performed with the indicated antibodies (*n* = 3). All data are expressed as the means ± S.D. * *p* < 0.05 versus the only-RANKL-treated group obtained through the unpaired Student’s *t*-test.

**Figure 5 pharmaceuticals-13-00468-f005:**
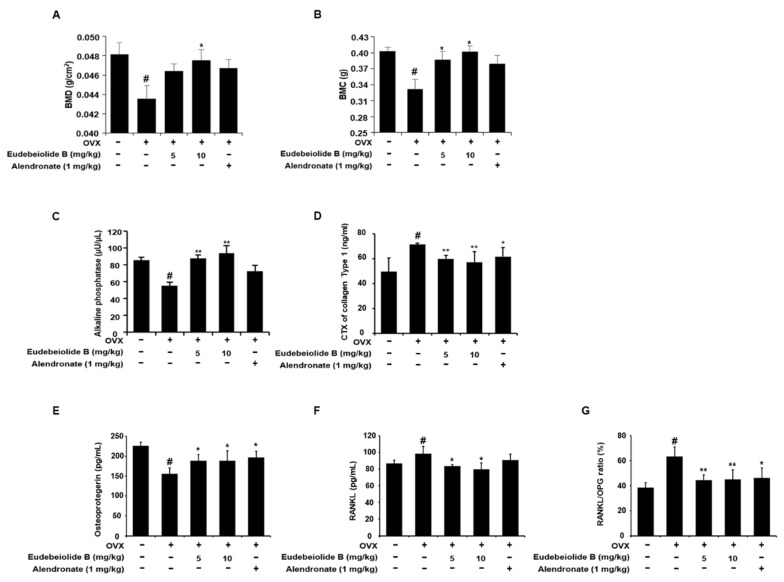
Bone mineral density BMD, bone mineral content BMC and biochemical serum analysis of the sham-operated, OVX + vehicle, OVX + eudebeiolide B-treated and OVX + alendronate-treated mice. (**A**) BMD and (**B**) BMC of the whole body were analyzed using DEXA (*n* = 6). The serum levels of (**C**) ALP, (**D**) c-terminal telopeptide of type 1 collagen (CTX), (**E**) osteoprotegerin (OPG) and (**F**) RANKL were determined by ELISA (*n* = 6). (**G**) RANKL/OPG ratio was calculated (*n* = 6). Data are expressed as the means ± S.D. # *p* < 0.05 versus the sham group obtained through the unpaired Student’s *t*-test; * *p* < 0.05, ** *p* < 0.01 versus the only-OVX-treated group obtained through one-way ANOVA followed by Dunnett’s test.

**Figure 6 pharmaceuticals-13-00468-f006:**
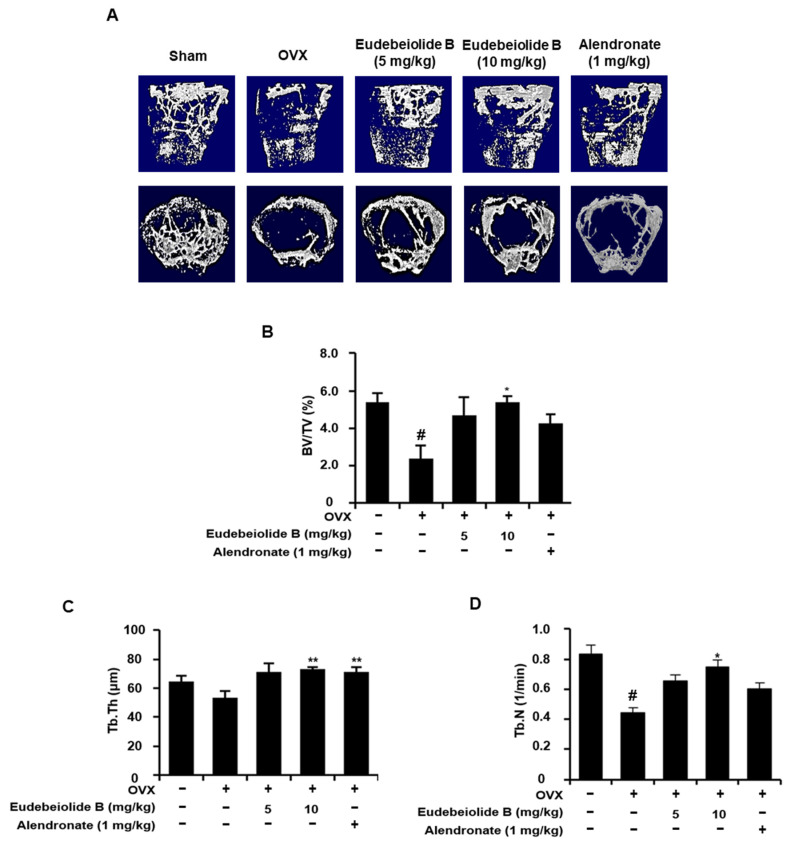
Micro-CT analysis of the proximal femurs from the eudebeiolide B- or alendronate-treated OVX mice. (**A**) The micro-CT images of longitudinal and transverse of the proximal femurs from the sham-operated, OVX + vehicle, OVX + eudebeiolide B-treated and OVX + alendronate-treated mice were obtained. (**B**) The bone volume/total volume (BV/TV), (**C**) trabecular thickness (Tb. Th.) and (**D**) trabecular number (Tb. N.) of the femurs were analyzed using micro-CT (*n* = 6). Data are expressed as the means ± S.D. # *p* < 0.05 versus the sham group obtained through the unpaired Student’s *t*-test; * *p* < 0.05, ** *p* < 0.01 versus the only-OVX-treated group obtained through one-way ANOVA followed by Dunnett’s test.

**Figure 7 pharmaceuticals-13-00468-f007:**
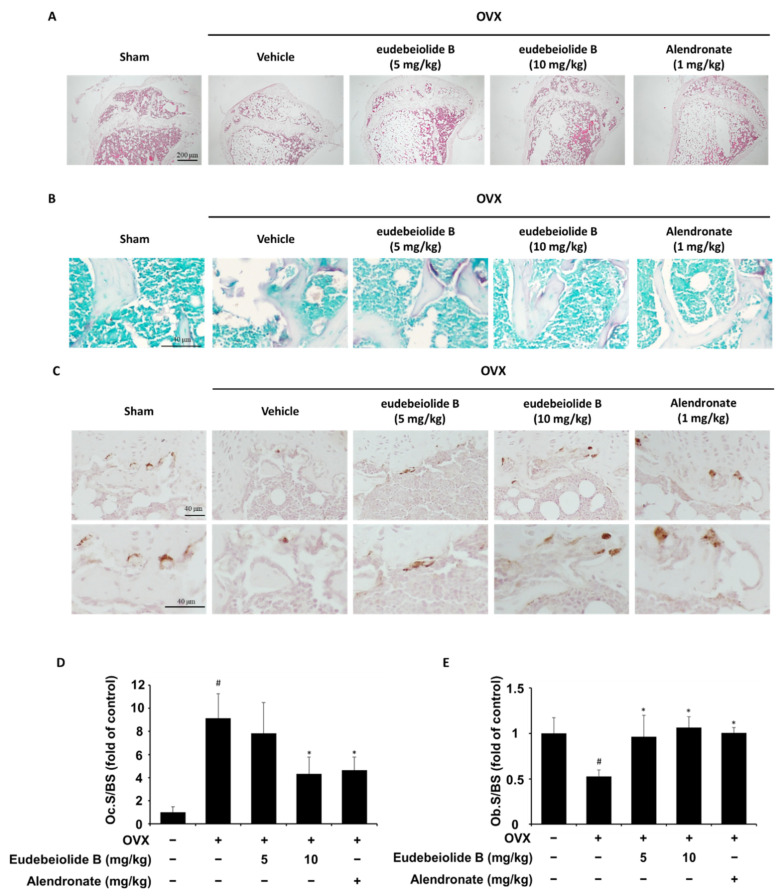
Histological analysis of the proximal tibia tissue sections from the eudebeiolide B- or alendronate-treated OVX mice. (**A**) H&E stains, (**B**) histochemical stains for TRAP and (**C**) immunohistochemical stains for ALP of the sham-operated, OVX + vehicle, OVX + eudebeiolide B-treated and OVX + alendronate-treated mice were obtained (*n* = 6). (**D**) Osteoclast surface/bone surface (Oc.S/BS) and (**E**) Osteoblast surface/bone surface (Ob.S/BS) were evaluated by HistoMorph software (*n* = 6). Data are expressed as the means ± S.D. # *p* < 0.05 versus the sham group obtained through the unpaired Student’s *t*-test; * *p* < 0.05 versus the only-OVX-treated group obtained through one-way ANOVA followed by Dunnett’s test.

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
