# Peer review of "Eudebeiolide B Inhibits Osteoclastogenesis and Prevents Ovariectomy-Induced Bone Loss by Regulating RANKL-Induced NF-κB, c-Fos and Calcium Signaling"

_pharmaceuticals, 2020, doi:10.3390/ph13120468_

Round 1

Reviewer 1 Report

Kim and colleagues investigate the effects of Eudebeiolide B on osteoclastogenesis and as a protective treatment against OVX-induced bone loss. Although the concept sustaining this study is both interesting and promising, using natural plant-derived compounds instead of pharmaceuticals therapeutic agents, the data presented in this manuscript and the way they are presented are not convincing. This paper is not yet ready for publication.

Figure 1 : In the description of the data, the fact that the cell viability has been made on BMM should be mentioned in the text not only in the legend.

Why the 1uM dose of Eudebeiolide B is missing on the panel D. It is perfectly understandable that the authors didn’t pursue this dose, but it should be explain.

The osteoblastic data are not convincing. Why have those 2 doses been chosen ? The red alizarin staining is not convincing at all.

The bone nodules should be illustrated if there is a real osteoblastic differentiation.

The authors should investigate some osteoblastic differentiation markers and specifically RANKL and OPG expressions.

Figures 2, 3 and 4

On the previous BMM culture for osteoclastic differentiation, the cells were cultured first with RANKL then with Eudebeiolide B. For those experiments, the culture conditions changed. There are a pretreatment with Eudebeiolide B then a treatment with RANKL. Basically the data presented with those western blots can’t explain what was observed on the previous cultures as the conditions are not the same. Moreover, the western blot data, their quantification and the statistical analysis are not at all convincing. What the authors described is not what can be seen on the illustrations.

Figure 5

In the description of the data the effect of the alendronate treatment is never described. What is the point to show them if they are neither described nor discussed?

BMD and BMC are not really histomorphometric data.

In the figure 5 panel A for example (but almost all the graphs and their statics analysis are like that), how the BMD the sham group is not significantly different from the OVX group ?

Could the authors explain the choice of the 2 doses ? Any links with the doses used in vitro ? It is stated that the treatment is orally administrated does that mean via the drinking water or otherwise ?

Figure 6 Are those bones really femurs ? All the images have to be commented. What is the different information the authors want to illustrate with the 2 first rows ? Is there a cortical bone phenotype that should be seen ?

Figure 7 : the analysis of the osteoclasts is not carried out on the appropriate region of interest. The osteoclasts are apparently counted right under the growth plate which is not the standard procedure. 

Interestingly, it seems that there is a lot of adipose tissues on the bone marrow space of those bones. Is there any differences between the different groups ?

Concerning the materials and methods sections, there are also quite some information missing.

The cell density is not always given (4.5, 4.8).

4.9. No washing has been made after the red alizarin staining ?

4.10. What are the concentrations of the antibodies both primary and secondary used ?

4.14. The microCT acquisition, reconstruction and analysis should be more developed. Filter used, rotation step, settings for the reconstruction, region of interest for the analysis.

The paper needs to be read again thoroughly, there some sentences that don’t make sense (for instance line 131) and quite a lot of misspellings (for instance 309).

The discussion section is not a real one. Can the authors speculate on how Eudebeiolide B is working on the cells ? Via a receptor, directly through the membrane ? This should be discussed.

In summary, in this version although the objective is very interesting there are too many inconsistencies, weird statistics, histomorphometric analysis not properly done, no real discussion, for the paper to be published.

Author Response

Thank you for review. Please check the attached file for the answer.

Reviewer 2 Report

In the manuscript “Eudebeiolide B inhibits osteoclastogenesis and prevents ovariectomy-induced bone loss by regulating RANKL-induced calcium signaling,” Kim et al investigated the effects of an eudesmane-type sesquiterpenoid compound eudebeiolide B on osteoblast differentiation, RANKL-induced osteoclastogenesis and ovariectomy-induced bone loss. While eudebeiolide B promotes osteoblast differentiation, RANKL-induced osteoclastogenesis was markedly reduced in eudebeiolide B-treated BMMs. Accordingly, oral administration of eudebeiolide B prevented estrogen deficiency-induced bone loss in mice. Mechanistically, eudebeiolide B suppress RANKL-induced calcium signaling in osteoclasts. Overall, this paper is presented logically and well-written, and experiments are thorough and well-controlled. However, the manuscript has several significant concerns as noted below. 

  1. In Figure S1, it is not clear why eudebeiolide B was selected among other eudebeiolide members because inhibitory activity of eudebeiolide C and D are as potent as that of eudebeiolide B, while eudebeiolide A, E, and G also show a suppressive activity.
  2. In Figure 1E and F, there is no explanation showing why treatment of eudebeiolide B promotes osteogenic differentiation. Eudebeiolide B-mediated inhibition of calcium signaling in osteoblasts may be responsible for this enhanced osteoblast differentiation? Expression of osteogenic marker genes should be assessed in eudebeiolide B-treated MC3T3-E1 cells.
  3. In line 109, calcium accumulation should be changed to mineralization.
  4. In line 131, remove “As shown in Figure 2”.
  5. In Figures 2-4, RANKL-induced expression of c-Fos and Nfatc1, phosphorylation of NF-kB/p65, and phosphorylation of PLCr, Btk, CaMKIV, and CREB were all reduced in eudebeiolide B-treated BMMs, suggesting that eudebeiolide B inhibits calcium signaling as well as other signaling pathways, including NF-kB/p65 and c-Fos pathways. The title should be corrected accordingly.
  6. In line 173, please change “morphometric” to “histomorphometric”.
  7. In Figure 5c, osteocalcin expression data is confusing because its expression mainly represents bone formation activity rather than bone remodeling activity. Please remove this data from the manuscript.
  8. In Figures 5-7, anti-resorptive activity of alendronate treatment is very mild in OVX-mice. Additionally, Figure 7B and D show a significant reduction in TRAP-positive osteoclast numbers in the metaphyseal area, which is a confusing data. It has been reported that alendronate treatment increases number of TRAP-positive osteoclasts because alendronate inhibits osteoclast resorption rather than differentiation.
  9. Figure 7A needs high-powered images showing trabecular bone structure.
  10. In Figure 5-7 legends, vehicle only should be indicated.

Author Response

Thanks for review. Please check our manuscript (attachment).

Reviewer 3 Report

In this manuscript, the authors mainly studied the role of Eudebeiolide B, a compound isolated from Salvia pleberia R.Br,  on osteoclast differentiation. With both in vitro and in vivo experiments, they demonstrated that Eudebeiolide B inhibits osteoclast differentiation and prevents OVX-induced bone loss by regulating RANKL-induced calcium signaling. While the main finding that Eudebeiolide B inhibits osteoclastogenesis is convincing, I have couple main concerns:

  • Both Osteocalcin and Alkaline phosphatase are osteoblast activity markers. In the OVX model (Fig 5), why OVX increase osteocalcin but decrease Alkaline phosphatase? Same question, how to explain the effect of Eudebeiolide B on these 2 markers?
  • The main reason of increased osteoclast formation in OVX is increased RANKL and/or decreased OPG by osteoblast due to lack of Estrogen. Could the authors explain how Eudebeiolide B regulates OPG and RANKL level in their mouse model (Fig 5)? Does Eudebeiolide B regulate OPG and RANKL by osteoblast in vitro? If this is true, what is the main mechanism by which Eudebeiolide B protect OVX-induced bone lose: inhibiting osteoclast differentiation by regulating RANKL-induced calcium signaling in osteoclast cells or regulating OPG/RANKL ratio in osteoblast?
  • The main mechanism by which bisphosphate (Alendronate) protect bone loss is that this compound deposits in the bone matrix and induces osteoclast apoptosis. How Alendronate regulate OPG?

Author Response

(The authors gave the same response as above.)

Round 2

Reviewer 1 Report

The author answered a lot of points although the quality of the bone phenotype analysis and some of the other data are still not convincing and not sufficient to allow the publication of the paper in this version.

The new comments on the authors responses are below in bold.

Point 3) The osteoblastic data are not convincing. Why have those 2 doses been chosen?

Response 3) Thanks for your review. In osteoblast experiment for ALP and alizarin red staining, eudebeiolide B was treated for 7 and 21 days. The 30 μM of eudebeiolide B showed cytotoxicity in 21 days culture. Thus, we selected 1 and 10 μM. Additionally, this research was focused on effect of eudebeiolide B on osteoclastogenesis and its effect on osteoblast differentiation was added in supplementary data. However, another reviewer suggested to move osteoblastic data in main manuscript because it is the most important effect of eudebiolide B.*

The cytotoxicity effect should be added in the manuscript.

Point 4) The red alizarin staining is not convincing at all. The bone nodules should be illustrated if there is a real osteoblastic differentiation.

Response 4) Thanks for your review. Generally, red alizarin staining result shows bone nodules. However, if excessive calcium accumulation is occurred, it is difficult to identify bone nodules from entire well image. We added some references.

Thank you for demonstrating that others show red alizarin staining. What was asked was pictures of the actual nodules such as the second panel of the second reference that the authors kindly provided (the one by Wu et al, from PloS one).

Point 5) The authors should investigate some osteoblastic differentiation markers and specifically RANKL and OPG expressions.

Response 5) Thanks for your review. We added effect of eudebeiolide B on Runx2 and Osterix osteoblastic differentiation marker genes in figure 1G. Instead of RANKL and OPG expression in MC3T3-E1 cells, we added serum level of RANKL and OPG in figure 5E and F.

Thank you for the addition, but you should check the RANKL and OPG expression in vitro

Figure 2, 3 and 4

Point 6) On the previous BMM culture for osteoclastic differentiation, the cells were cultured first with RANKL then with Eudebeiolide B. For those experiments, the culture conditions changed. There are a pretreatment with Eudebeiolide B then a treatment with RANKL. Basically the data presented with those western blots can’t explain what was observed on the previous cultures as the conditions are not the same.

Response 6) Thanks for your review. Basically your comment about culture condition is right. However, many researches used the same method for evaluating osteoclast differentiation and signaling molecule. We added references. Osteoclast differentiation takes more than 4 days culture period. However, accessing signaling molecules takes an 5, 15, 30 and 60 min incubation time. And pretreatment of compounds could show more significant effect of treatment. For that reason, there are differences between evaluating osteoclast differentiation and signaling molecules protocols.

Sorry, but although the experimental difficulties are perfectly understandable, these data are not comparable. The authors should hence only present the pretreatment set up, no matter if some others have done the same. This is not rigorous as presented here.

Point 7) Moreover, the western blot data, their quantification and the statistical analysis are not at all convincing. What the authors described is not what can be seen on the illustrations.

Response 7) Thanks for your review. Our research was carried out by a defined methods, and similar research results were previously published. Also, the analysis method was clearly carried out without modification by the program (image J and GraphPad Prism 5).

The way the western blot was analyzed was not questioned. The problem is that more than once, the thickness of the western blot band is not in accordance with the quantification.

The list of antibodies and supplier would be nice to be added.

And the sequences of the primers used for the qPCR 

Figure 5

Point 8) In the description of the data the effect of the alendronate treatment is never described. What is the point to show them if they are neither described nor discussed?

Response 8) Thanks for your comment. Alendronate was used as a positive control in this study. We added this information in section 2.4 and 4.12.

Thanks for the addition. The data should be at least discussed.

Point 9) BMD and BMC are not really histomorphometric data. In the figure 5 panel A for example (but almost all the graphs and their statics analysis are like that), how the BMD the sham group is not significantly different from the OVX group?

Response 9) Thanks for your review. We revised it following your comment in figure legend of figure 5. And there are significant differences between sham and OVX group. We revised and added it in figure 5, 6 and 7.

It is stil not clear how the data were statistically analyzed. Actually, it is even more strange. The analysis should be checked. It is unclear how 2 differents reference group could be used with a Tukey test.

Point 10) Could the authors explain the choice of the 2 doses? Any links with the doses used in vitro? It is stated that the treatment is orally administrated does that mean via the drinking water or otherwise?

Response 10) Thanks for your comment. Basically, it is impossible to choose in vivo concentration based on the in vitro result. The compound goes through various metabolic pathways and could react with various biological substances. Thus, we determined in vivo dosages based on previous study. The previous study of Salvia plebeian R. Br. extract, we treated extract up to 500 mg/kg. And we obtained 24.5 mg of eudebeiolide B from 2.6 kg of ethanol extract. It means 500 mg/kg dosage of extract is containing almost 5 mg/kg of edebeiolide B. Furthermore, we added 10 mg/kg of eudebeiolide B to show significant therapeutic effect because extract contains other active compounds. Finally, oral gavage was used for oral administration.

Thank you for the explanation. This should be added. But you cannot use ‘oral administration’ when it is in really gavage. Please correct.

Figure 6

Point 11) Are those bones really femurs? All the images have to be commented. What is the different information the authors want to illustrate with the 2 first rows? Is there a cortical bone phenotype that should be seen?

Response 11) Thanks for your review. All images were obtained from distal femur. The images of 2 first rows are longitudinal and transverse sections of distal femur. We revised and added this information in figure legend of figure 6. We presented these images to show trabecular bone status. We are sorry that we don’t have cortical bone data. Instead of that 3-dimensional image of femur could show it phenotype.

The third lane should be removed if cortical bone is not commented.

Figure 7

Point 12) The analysis of the osteoclasts is not carried out on the appropriate region of interest. The osteoclasts are apparently counted right under the growth plate which is not the standard procedure.

Response 12) Thanks for your review. We followed some references and added references.

Thanks for the references. But instead of showing references going your way, you should have used the international accepted and followed rules of bone analysis, meaning Parfitt et al paper and the ASBMR update paper (Parfitt et al. 1987; Dempster et al. 2013).

Materials and methods sections

Point 15) 4.9. No washing has been made after the red alizarin staining?

Response 15) Thanks for your review. Before fixation and staining processes, cells were washed with PBS. We revised and added it in manuscript.

But there was no washing before the dissolution ?

Point 18) The paper needs to be read again thoroughly, there some sentences that don’t make sense (for instance line 131) and quite a lot of misspellings (for instance 309).

Response 18) Thanks for your comment. We revised it.

Misspelling line 388.

Reconsider the sentence line 112

Point 19) The discussion section is not a real one. Can the authors speculate on how Eudebeiolide B is working on the cells ? Via a receptor, directly through the membrane ? This should be discussed.

Response 19) Thanks for your review. We revised discussion following your comment. We expect that eudebeiolide B exerts its activity through directly inhibiting specific signaling molecules. Binding of RANKL to RANK activates MAPKs and NF-kB signaling through TRAF6 and calcium signaling. If Eudebeiolide B inhibited receptor directly, downstream signaling pathways include MAPKs, NF-kB and calcium signaling were inhibited by eudebeiolide B treatment. But MAPKs signaling was not affected by eudebeiolide B treatment. These results imply eudebeiolide B exerts its activity through directly inhibiting specific signaling molecules.

So you implying that Eudebeiolide B goes through the membrane and acts on the signaling molecule ? If so this should be added.

Author Response

The author answered a lot of points although the quality of the bone phenotype analysis and some of the other data are still not convincing and not sufficient to allow the publication of the paper in this version.

The new comments on the authors responses are below in bold.

Point 3) The osteoblastic data are not convincing. Why have those 2 doses been chosen?

Response 3) Thanks for your review. In osteoblast experiment for ALP and alizarin red staining, eudebeiolide B was treated for 7 and 21 days. The 30 μM of eudebeiolide B showed cytotoxicity in 21 days culture. Thus, we selected 1 and 10 μM. Additionally, this research was focused on effect of eudebeiolide B on osteoclastogenesis and its effect on osteoblast differentiation was added in supplementary data. However, another reviewer suggested to move osteoblastic data in main manuscript because it is the most important effect of eudebiolide B.*

The cytotoxicity effect should be added in the manuscript.

Response) Thanks for your comment. We added it in supplementary data.

Point 4) The red alizarin staining is not convincing at all. The bone nodules should be illustrated if there is a real osteoblastic differentiation.

Response 4) Thanks for your review. Generally, red alizarin staining result shows bone nodules. However, if excessive calcium accumulation is occurred, it is difficult to identify bone nodules from entire well image. We added some references.

Thank you for demonstrating that others show red alizarin staining. What was asked was pictures of the actual nodules such as the second panel of the second reference that the authors kindly provided (the one by Wu et al, from PloS one).

Point 5) The authors should investigate some osteoblastic differentiation markers and specifically RANKL and OPG expressions.

Response 5) Thanks for your review. We added effect of eudebeiolide B on Runx2 and Osterix osteoblastic differentiation marker genes in figure 1G. Instead of RANKL and OPG expression in MC3T3-E1 cells, we added serum level of RANKL and OPG in figure 5E and F.

Thank you for the addition, but you should check the RANKL and OPG expression in vitro

Response) Thanks for your review. We added it in Figure 1G.

Figure 2, 3 and 4

Point 6) On the previous BMM culture for osteoclastic differentiation, the cells were cultured first with RANKL then with Eudebeiolide B. For those experiments, the culture conditions changed. There are a pretreatment with Eudebeiolide B then a treatment with RANKL. Basically the data presented with those western blots can’t explain what was observed on the previous cultures as the conditions are not the same.

Response 6) Thanks for your review. Basically your comment about culture condition is right. However, many researches used the same method for evaluating osteoclast differentiation and signaling molecule. We added references. Osteoclast differentiation takes more than 4 days culture period. However, accessing signaling molecules takes an 5, 15, 30 and 60 min incubation time. And pretreatment of compounds could show more significant effect of treatment. For that reason, there are differences between evaluating osteoclast differentiation and signaling molecules protocols.

Sorry, but although the experimental difficulties are perfectly understandable, these data are not comparable. The authors should hence only present the pretreatment set up, no matter if some others have done the same. This is not rigorous as presented here.

Response) Thanks for your review. We examined effect of eudebeiolide B pretreatment on osteoclastogenesis and replaced previous data.

Point 7) Moreover, the western blot data, their quantification and the statistical analysis are not at all convincing. What the authors described is not what can be seen on the illustrations.

Response 7) Thanks for your review. Our research was carried out by a defined methods, and similar research results were previously published. Also, the analysis method was clearly carried out without modification by the program (image J and GraphPad Prism 5).

The way the western blot was analyzed was not questioned. The problem is that more than once, the thickness of the western blot band is not in accordance with the quantification.

The list of antibodies and supplier would be nice to be added.

And the sequences of the primers used for the qPCR 

Response) Thanks for your review. We quantified all band density and replaced P38, JNK and Btk data. We used taqman gene assay primer from applied biosystems company in this study. However, they don’t provide primer sequence.

Figure 5

Point 8) In the description of the data the effect of the alendronate treatment is never described. What is the point to show them if they are neither described nor discussed?

Response 8) Thanks for your comment. Alendronate was used as a positive control in this study. We added this information in section 2.4 and 4.12.

Thanks for the addition. The data should be at least discussed.

Point 9) BMD and BMC are not really histomorphometric data. In the figure 5 panel A for example (but almost all the graphs and their statics analysis are like that), how the BMD the sham group is not significantly different from the OVX group?

Response 9) Thanks for your review. We revised it following your comment in figure legend of figure 5. And there are significant differences between sham and OVX group. We revised and added it in figure 5, 6 and 7.

It is still not clear how the data were statistically analyzed. Actually, it is even more strange. The analysis should be checked. It is unclear how 2 differents reference group could be used with a Tukey test.

Response) Thanks for your review and we are sorry for your confusing. We used Dunnett’s test to compare with only OVX group and unpaired student t test to compare Sham and only OVX group. We revised statistical analysis section in material method and added statistical analysis method in each figure legend.

Point 10) Could the authors explain the choice of the 2 doses? Any links with the doses used in vitro? It is stated that the treatment is orally administrated does that mean via the drinking water or otherwise?

Response 10) Thanks for your comment. Basically, it is impossible to choose in vivo concentration based on the in vitro result. The compound goes through various metabolic pathways and could react with various biological substances. Thus, we determined in vivo dosages based on previous study. The previous study of Salvia plebeian R. Br. extract, we treated extract up to 500 mg/kg. And we obtained 24.5 mg of eudebeiolide B from 2.6 kg of ethanol extract. It means 500 mg/kg dosage of extract is containing almost 5 mg/kg of edebeiolide B. Furthermore, we added 10 mg/kg of eudebeiolide B to show significant therapeutic effect because extract contains other active compounds. Finally, oral gavage was used for oral administration.

Thank you for the explanation. This should be added. But you cannot use ‘oral administration’ when it is in really gavage. Please correct.

Response) Thanks for your review. We added explanation about eudebeiolide B dosage in section 2.4. We replaced oral administration with intragastric injection.

Figure 6

Point 11) Are those bones really femurs? All the images have to be commented. What is the different information the authors want to illustrate with the 2 first rows? Is there a cortical bone phenotype that should be seen?

Response 11) Thanks for your review. All images were obtained from distal femur. The images of 2 first rows are longitudinal and transverse sections of distal femur. We revised and added this information in figure legend of figure 6. We presented these images to show trabecular bone status. We are sorry that we don’t have cortical bone data. Instead of that 3-dimensional image of femur could show it phenotype.

The third lane should be removed if cortical bone is not commented.

Response) Thanks for your comment. We removed it.

Figure 7

Point 12) The analysis of the osteoclasts is not carried out on the appropriate region of interest. The osteoclasts are apparently counted right under the growth plate which is not the standard procedure.

Response 12) Thanks for your review. We followed some references and added references.

Thanks for the references. But instead of showing references going your way, you should have used the international accepted and followed rules of bone analysis, meaning Parfitt et al paper and the ASBMR update paper (Parfitt et al. 1987; Dempster et al. 2013).

 Response) Thanks for your review. We analyzed osteoclast area in cancellous bone according to ASBMR update paper.

Materials and methods sections

Point 15) 4.9. No washing has been made after the red alizarin staining?

Response 15) Thanks for your review. Before fixation and staining processes, cells were washed with PBS. We revised and added it in manuscript.

But there was no washing before the dissolution ?

Response) Thanks for your comment. The cells were washed with distilled water before dissolution. We revised manuscript.

Point 18) The paper needs to be read again thoroughly, there some sentences that don’t make sense (for instance line 131) and quite a lot of misspellings (for instance 309).

Response 18) Thanks for your comment. We revised it.

Misspelling line 388.

Reconsider the sentence line 112

Response) Thanks for your comment. We revised manuscript.

Point 19) The discussion section is not a real one. Can the authors speculate on how Eudebeiolide B is working on the cells ? Via a receptor, directly through the membrane ? This should be discussed.

Response 19) Thanks for your review. We revised discussion following your comment. We expect that eudebeiolide B exerts its activity through directly inhibiting specific signaling molecules. Binding of RANKL to RANK activates MAPKs and NF-kB signaling through TRAF6 and calcium signaling. If Eudebeiolide B inhibited receptor directly, downstream signaling pathways include MAPKs, NF-kB and calcium signaling were inhibited by eudebeiolide B treatment. But MAPKs signaling was not affected by eudebeiolide B treatment. These results imply eudebeiolide B exerts its activity through directly inhibiting specific signaling molecules.

So you implying that Eudebeiolide B goes through the membrane and acts on the signaling molecule ? If so this should be added.

Response) Thanks for your comment. We added it in discussion section.

Reviewer 2 Report

No further comments

Author Response

Thanks for your review.

Reviewer 3 Report

After carefully review their response and manuscript, I need mention couple points:

  • Both osteocalcin and alkaline phosphatase are specific osteoblast activity (osteoblastic bone formation) markers and osteoclast does not express either of them. I am so surprised to see that “It has already been reported that osteoclast has osteocalcin and Alkaline phosphatase as major biomarkers.” Although they cited 1 reference which might came from their lab to show increased osteocalcin and decreased alkaline phosphatase after OVX, it does not make sense. The authors removed Osteocalcin data in the revised version without mentioning.
  • The authors did not address my question: Does Eudebeiolide B regulate RANKL and OPG expression in osteoblast? If this is true, what is the main mechanism by which Eudebeiolide B protect bone loss in OVX? directly inhibits NF-κB, c-Fos and calcium signaling in osteoclast or regulate RANKL OPG expression by osteoblast in OVX?

Author Response

Thanks for your review.

We attached revision file.

Round 3

Reviewer 3 Report

They addressed mu concerns and it is OK to accept for publication.